# Comparative Analysis of Performance between Multimodal Implementation of Chatbot Based on News Classification Data Using Categories

**Prasnurzaki Anki [1]** , **Alhadi Bustamam [1,2,]*** **and Rinaldi Anwar Buyung [3,4]**

1   Department of Mathematics, Faculty of Mathematics and Natural Science, Universitas Indonesia, Depok 16424, Indonesia; prasnurzaki.anki@sci.ui.ac.id

2   Data Science Centre, Faculty of Mathematics and Natural Science, Universitas Indonesia, Depok 16424, Indonesia

3   Department of Management, Faculty of Economics and Business, Universitas Indonesia, Jakarta 10430, Indonesia; rinaldiFSAI@gmail.com

4   Global Risk Management, Jakarta 12540, Indonesia

*   Correspondence: alhadi@sci.ui.ac.id

**Abstract:** In the modern era, the implementation of chatbot can be used in various fields of science. This research will focus on the application of sentence classification using the News Aggregator Dataset that is used to test the model against the categories determined to create the chatbot program. The results of the chatbot program trial by multimodal implementation applied four models (GRU, Bi-GRU, 1D CNN, 1D CNN Transpose) with six variations of parameters to produce the best results from the entire trial. The best test results from this research for the chatbot program using the 1D CNN Transpose model are the best models with detailed characteristics in this research, which produces an accuracy value of 0.9919. The test results on both types of chatbot are expected to produce sentence prediction results and precise and accurate detection results. The stages in making the program are explained in detail; therefore, it is hoped that program users can understand not only how to use the program by entering an input and receiving program output results that are explained in more detail in each sub-topic of this study.

**Keywords:** chatbot; GRU; Bi-GRU; 1D CNN; 1D CNN transpose

## 1. Introduction

An automated system that assists users in answering their questions is called a chatbot. To increase the level of customer satisfaction in the business world, chatbots can be used to provide a better way to connect with customers In this case, it will be beneficial for customers to get answers to their questions using a better and more convenient way, without waiting for a call or frequent email [1]. The impact on daily activities has been given by artificial intelligence (AI), one of which is in the form of developing chatbots with various functions, which will be explained in the following references:

- The implementation of a chatbot during an ongoing crisis, which uses the 2020 COVID-19 pandemic as a case study on research [2].
- In another study, the chatbot architecture was investigated as a whole and provided the appropriate metamodel and rules used for mapping between the proposed and the two natural metamodels, commonly used language understanding (NLU) [3].
- Another reference discusses various chatbot development challenges that are answered through an agent-based framework for chatbot development called EREBOT, which has been tested in the context of preserving physical balance in times of social confinement (due to the ongoing pandemic) [4].

Chatbot is a form of intelligence based on AI. The hope of making a chatbot is that it can imitate conversations made by human agents. Service fees are proven to be reduced

by a chatbot because it can handle many customers simultaneously, and it is becoming very common among them [5]. Moreover, it is hoped that a chatbot can increase the level of customer satisfaction in the business world because it can save more time. In addition, a chatbot is very helpful for customers to obtain information that previously had to be answered manually by humans. Using a chatbot is now more practical.

The form that has been filled out by the user will be used as a retrieval model, where it contains answers from users that come from several forms based on matches obtained from users and chatbots. The human hand-code serves as the knowledge that is used in the chatbot. It is time-consuming and difficult to build knowledge about chatbots; therefore, the automatic knowledge extraction mechanism is very important, because the development of various forms of chatbots can be achieved through this [6]. The use of models that can improve chatbot performance in answering questions automatically can be considered, where we can compare two different models and determine which model has more influence on chatbot performance and which model is more suitable.

One form of recent technological advances that is very helpful in the development of new virtual assistants to be efficient is AI. Therefore, the right use of AI is a challenge in this modern era. In this study, we will analyse how technological advances in new chatbots impact customer support in the future. Furthermore, it is hoped that in the future, complex tasks can be increased by chatbots through technological innovations that will adapt to the times [7]. Natural language processing (NLP) is the understanding of language through simulating human capabilities derived from the mechanisms that support computer machines [8]. In handling the sparse bag of words matrix, one of the techniques used is embedding. One of them is the application of suffix features in artificial neural networks by inserting words that are not visible from their spelling or morphology by constructing their insertion [9]. Another area of shaping deep learning can have a major impact on future experiments; this is usually the case with NLP [10]. The type of network based on a recurrent neural network (RNN) is the gated recurrent unit (GRU); the GRU can also study the sentence aspect. Based on [11], RNN sequence modelling capabilities can make bi-directional gated recurrent units (Bi-GRUs) selected to convert word insertion vectors into underlying representations. According to previous research in [12], convolutional neural networks (CNNs) can be used to perform feature extraction by considering the context information as a whole sentence that serves to build word embedding with context sensitivity or integrate all the information in the sentence to compose a sentence embedding step. Based on the development of existing models, according to other studies, [12] is the first to extend deconvolution (that is, transposed convolution) to create word embedding. Based on these various references, the application of the CNN model to the adjustment of the data form used, as well as the expansion of deconvolution with transposed convolution, 1D CNN and 1D CNN Transpose were selected as the other two models to be tested. Based on these various references, it can be concluded that the selection of models to be tested for the chatbot program is GRU, Bi-GRU, 1D CNN and 1D CNN Transpose.

When deciding to start building a chatbot program, the things to consider from the implementation question and answer system records a number of questions originating from problems that have been experienced by the consumers in general [6], they are related to various obstacles in data storage, a limited customer service that does not operate for a full day and programs that cannot operate effectively. The motivation of this research is to speed up the process of obtaining answers to a question. The objectives of this research are to obtain accurate and fast results using the work of the chatbot program in answering problems; it is hoped that it will reduce the working hours of customer service and increase consumer confidence in the company with this technology.

Based on the results of discussions from several chatbot backgrounds, consumer problems are generally conveyed through several recorded questions related to problems, such as the limited number of hours of customer service and data storage, which is the research question of this study. Thus, a chatbot program that will be built will be able to

answer the questions given by consumers and can optimise the results of these services needs to be provided. In relation to this, the current study will find a way to acquire these research goals, such as speeding up the search for answers and improving performance efficiency using the modelling theory that will be discussed.

## 2. Materials and Methods

A simple robot within which there is a program that can answer questions from users is called a chatbot. The program contains data that can produce answers to questions that are submitted by users. Thus, a resulting semantic question-and-answer system was developed, and the resulting words are not certain to be in the form of questions [13]. The application of chatbots in a question-and-answer system is expected to answer these challenges.

### 2.1. Steps in Creating a Chatbot

The four steps in creating a chatbot include selecting the dataset that will be used in the chatbot program, inputting question and answer system data, compiling the chatbot program, and finally evaluating the output.

#### 2.1.1. Selection of Datasets to Test Classification Models on the Chatbot Program

Based on sources from reference [14], News Aggregator Dataset (data for November 2016) is used to test the classification model against predetermined categories. The database is set by the UCI Machine Learning Repository, which will be useful to test whether the classification of news from various categories using data from the dataset can be tested in the chatbot program or not. It is used in the empirical analysis of machine learning algorithms in the machine learning community.

#### 2.1.2. Question and Answer System Data Input

In performing input from program users, a file containing input sentences that will be entered by program users into the program through a dialog box that is displayed, based on the input sentences, will be processed through a classification model against a predetermined category; once identified, the input sentence will be classified into a category.

#### 2.1.3. Compiling the Chatbot Program

In the preparation of the building chatbot program, there are several things that need to be considered; the selection of a classification model for the categories will be tested including GRU, Bi-GRU, 1D CNN and 1D CNN Transpose. From all of that, the best model will be determined, which will be applied as a model in the chatbot program.

#### 2.1.4. Output Evaluation

The model is evaluated to determine whether the model that has been applied gives accurate results. In the multimodal implementation of a chatbot, it will be tested to confirm whether the model selected as the model used has met expectations to be able to categorise a sentence.

### 2.2. GRU Model

The RNN variant has a higher value in its ability to take steps of sequential learning variables than other deep learning approaches; an example of a sequential learning variable is text size [15]. The results in [16] have reported the advantages of the algorithm used in this study for variable length on input–output. RNN applications (GRU and LSTM) have been widely used in the fields of speech recognition, machine translation NLP, etc. Based on [17–19], the feasibility of this method has been tested on text classification problems. NLP and supervised machine learning tasks are used for text classification problems, to find out knowledge and information from accident reports that have a function in order to prevent accidents in the future when working on a construction project using datasets

based on the classification of various safety measures; it is fundamentally important for this to be tested in news classifications in our research whether it can be tested or not [20].

Based on reference [21], there are two gates in the algorithm with memory cells; the two gates are called the renewal gate and the reset gate. A new type of hidden unit is called a GRU.

Time $t$ binds each process state in the GRU. This process starts when the time step values $t$, $x_t$ and the hidden state are entered in the step $h_{t-1}$, which were previously calculated as update gates and reset gates. Furthermore, as a candidate state for the calculation of the current time step, they use the reset gate value. The renewal gate value and candidate state with time step $t$ are used in the last step in the GRU memory cell process, which is indicated by the hidden state calculation. $\varnothing$ is the Ridge Function, which is a multivariate function that works on linear combinations resulting from the input variables. $\sigma$ and @ are sigmoid functions, which are a form of activation function. $Wx_r$, $Wx_z$, $Wx_c$, $Wx_h$, $Wh_r$, $Wh_z$, $Wh_c$, $Wh_h$, $b_r$, $b_z$, $b_c$ and $b_h$ are the weights and biases associated with each gate.

The GRU process, in detail, is regulated in the following equation [21]:

1.　Gate $R(t)$ is the reset, set whether the hidden status of the previous information is ignored or not.

$$R(t) = \sigma(Wx_r * h(t-1) + Wh_r * X(t) + b_r) \tag{1}$$

The $Z(t)$ gate is updated; it is performed by remembering the long-term information and deciding which value to pass to the other time step block.

$$Z(t) = \sigma(Wx_z * h(t-1) + Wh_z * X(t) + b_z). \tag{2}$$

2.　In calculating the hidden state, one of the inputs uses the reset gate value, the current input and also the hidden state that has been successfully lowered to the GRU candidate status ($\hat{C}(t)$).

$$\hat{C}(t) = \varnothing(Wx_c * X(t) + Wh_c * (r \odot h(t-1)) + b_c \tag{3}$$

3.　The update gate value $h(t)$ is used in the GRU hidden state; it is used to determine the value of the current time step or the value of the previous hidden state, all of which are candidates for the hidden state value of this time step.

$$h(t) = \sigma(Z_t * h(t-1) + (1 - Z_t) * \hat{C}(t) + b_h). \tag{4}$$

4.　The weights and biases involved in the neural network equation are the result of the GRU output layer ($Ot$).
$$Ot = W_y * h(t) + by, \tag{5}$$

### 2.3. Bi-GRU Model

RNNs take words that sequentially interpret the document. RNNs are remote dependent; therefore, they are very difficult to train. To overcome this problem, RNNs introduced several variants, such as GRU and LSTM [17,22]. Furthermore, the input sequence with the renewal gate and reset gate will be controlled by the GRU network [23]. Based on reference [11], the word insertion factor into the underlying representation can be changed by selecting Bi-GRU from the RNN sequence modelling capability. The calculation of one GRU layer at time step $i$ is as follows:

$$z_i = \sigma(W^z x_i + U^z h_{t-1}), \tag{6}$$

$$r_i = \sigma(W^r x_i + Urz h_{t-1}), \tag{7}$$

$$h_i = (1 - z_i) \odot h_{i-1} + z_t \odot g_i \tag{8}$$

$$g_i = f\left(W^h x_i + U^h (r_i \odot h_{i-1})\right), \tag{9}$$

where $x_i$ is the element *input sequence*; $g_i$ is the *output* calculated; $z_i$ is the *update gate*; $r_i$ is the *forget gate*; $\sigma, f$ and @ each show an activation function sigmoid; $h_i$ is the *hidden gate* tanh activation function and element multiplication. In the two-way GRU model, *hidden states* of forward and backward propagation are combined as the following token representation $h_i^\beta$:

$$h_i^\beta = [\overrightarrow{GRU}(v_i); \overleftarrow{GRU}(v_i) \tag{10}$$

Here, the equation for $v_i$ in the form of the multiplication operation of the weight l-th layer representation and the token representation divided by $s = \sum_{l=1}^L 2^l$ is as follows:

$$v_i = \sum_{l=1}^L \frac{a_l h_i^l}{s}, \tag{11}$$

where $a_i$ is the weight *l*-th layer representation. Then, $a_l = 2^i$ was selected in this research [11].

Bi-GRU works in parallel, which uses two GRU layers that can work side by side. The output generated from the Bi-GRU model is a combination of the two outputs generated through the GRU forward and backward sequence processes [24,25]. Based on the explanation of the previous sentence, which states that one of the models tested in this study, Bi-GRU, works in parallel, parallel work based on parallel computing is a type of computing that can perform various process calculations that are carried out at the same time [26].

Unidirectional GRUs cannot handle many speech samples of different durations well. They can only shorten the duration of the speech sample and can even ignore the information behind the speech sequence, resulting in the loss of some speech features. Reminded of the importance of information order, the Bi-GRU model based on [25] can take advantage of the previous and subsequent information, because the sound expansion processing is more suitable to use it. The form has a shorter phonetic duration and can be effectively extended using it. In Figure 1 based on [27], shows the structure of the Bi-GRU.

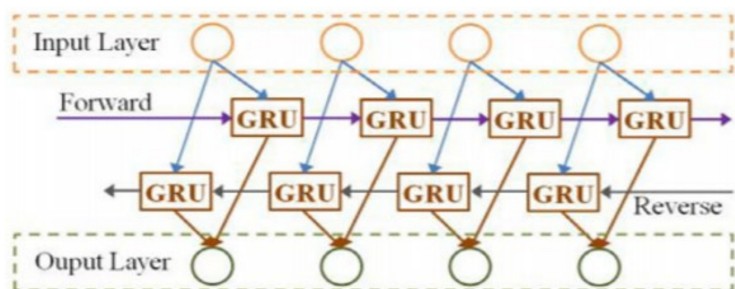

**Figure 1.** Structure of the *bi-directional gated recurrent unit* (Bi-GRU) model [24].

*2.4. D CNN Model*

In the processing of machine learning tasks and natural language, the problem of text classification will often be used, which will use the 1D CNN model for text data processing. The importance of discussing the text classification used in the 1D CNN model is due to the ability of the CNN model to be able to process text data in the form of one dimension. It is hoped that this model can be tested properly to be applied in the chatbot program that will be studied in this study. Several predictive categories can be classified from text classification, such as understanding people's sentiments on social media, detecting ham and spam emails and auto-tagging customer queries and topic prediction. In the research reference, the sentences are encoded using the encoding that is used as input for the CNN model [28,29].

Text mining extensively uses NLP in processing text and speech [30]. Minimal preprocessing is used from multilayer perceptrons, where CNN is a network class of feedforward neural. Initially, image classification and computer vision problems were developed by CNN. CNN is generally used in image processing, but in this research, we focused on developing CNN, which will be a model in NLP, namely text processing, by changing the dimensions, which are generally 2 dimensions and 3 dimensions on CNN, into 1 dimension. However, now there are many problems that can be solved using NLP, where the form of a one-dimensional (1D) array text representation in the NLP task is because the text form that can determine a data contains information, the task requires a CNN form in the form of one-dimensional text data. The 1D circuits and the convolution operations belong to the CNN architecture. Therefore, 1D convolution is a very important process from the image processing stage of area neural networks and also techniques in other fields. The word order $w_{(1:n)} = w_1 \ldots w_n$ is one way of thinking about this operation, where each word corresponds to its input vector of dimension $d$ and the other components act as convolutions for the kernel input word or convolution filter. Now slide to size $k$ of the kernel to input all the words by multiplying the input vector by each kernel value and also by the number of overlapping values. Between the set of input insertion vectors in a given window and the weight vector $u$, there is a point product, where the non-linear activation function $g$ often follows [31].

Considering the window of words $w_1 \ldots w_{(i+k)}$, the combined vector of the $i$-th window becomes $x_i$:

$$x_i = [w_i, w_{i+1}, \ldots \ldots, w_{i+k}] \in R^{(k*d)} \tag{12}$$

Each window will apply a convolution filter, where a scalar value of $r_i$ will be generated in each of the $i$-th windows.

$$r_i = g(x_i, u) \in R \tag{13}$$

The filter, $u_1, \ldots, u_l$, is more widely used in practice, which can then represent a vector multiplied by a matrix $U$ and by adding a bias term $b$:

$$r_i = g(x_i.U + b). \tag{14}$$

With $r_i \in R^l$, $x_i \in R^{(k*d)}$, $U \in R^{(k.d*l)}$ and $b \in R^l$.

An example of this 1D CNN model is presented in Figure 2 (based on the reference image from the reference [31]):

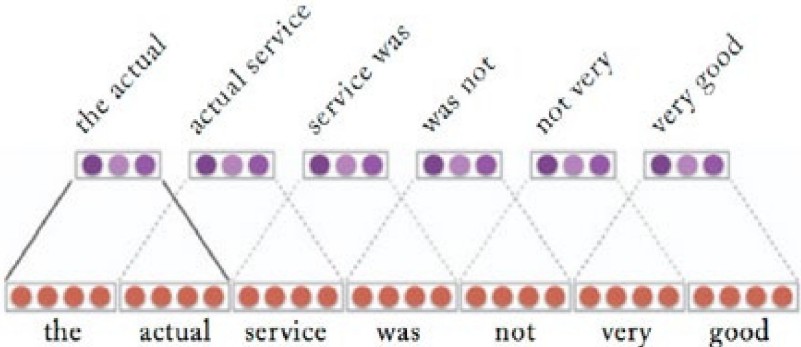

**Figure 2.** A convolutional sentence in vector merge notation [31].

### 2.5. D CNN Transpose Model

One layer deconvolution with $1 \times 1$ is the filter size and $1 \times 1$ is the deconvolution operation that will focus on the processing text data; it is equivalent to the $1 \times 1$ convolution operation. One by one, the convolution operation was first introduced by [32] and that

will be developed further on GoogLeNet [33]. When 1 is the step, then the formula for the convolution or deconvolution operations $1 \times 1$ is as follows:

$$\hat{H}_i = w_i H, \tag{15}$$

where the real number is $w_i$, representing the *i*-th filter, $H$. Next, from the previous layer, a feature map is displayed, and the size $H$ is equal to $\hat{H}_i$.

$l_w$, $l_h$, $l$ is a form of input tensor, where the special dimension of the feature map is represented by $l_w$, $l_h$ and the number of feature maps is denoted by *I*. After the tensor $(l_w, l_h, l)$ is fed into the convolution or deconvolution layer $1 \times 1$ with filter $\hat{I}$, then $l_w$, $l_h$, $\hat{I}$ will be formed from the output tensor of layer $1 \times 1$. Thus, based on this exposure, the dimensions in the filter chamber can be changed using a convolution layer or a $1 \times 1$ deconvolution layer. Furthermore, the statement that fits the description is when $\hat{I} > l$, then the $1 \times 1$ filter can increase the dimensions, but when $\hat{I} < l$, this can reduce the dimensions $(l_w, l_h, \hat{I})$.

From Equation (15), it is completely linear for the $1 \times 1$ convolution or deconvolution operation; besides that, there is an addition to the non-linear RELU activation layer that exists after the $1 \times 1$ deconvolution layer, which is a form of application of the workings of processing data in one-dimensional form in the form of text data [33].

For the application of an example of how to work between convolutional and deconvolutional on a data dimension, it will be shown in Figure 3.

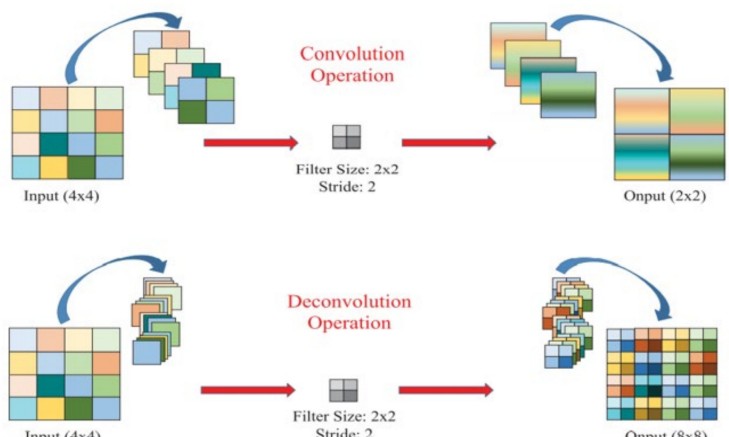

**Figure 3.** Illustration of convolution and deconvolution operations [34].

*2.6. Data Implementation in Python Programs*

Systematically, a program plan will be compiled for data implementation to Python with Jupyter Notebook Software, as follows:

(a)    The software to be used must be selected

Things that need to be considered in choosing the software used include whether the data processing of the selected data can run well to be processed in the program, the performance of the software in data processing. In addition, it is important to check the availability of support for the preparation of program attributes needed to be able to check readiness in making programs. Therefore, the selection of the Jupyter Notebook software has met the various requirements mentioned above.

(b)    Selecting the chatbot model and program structure

Models that match the characteristics of the data can affect program performance; it is because the main factor in determining the model can be influenced by the level of accuracy of a program. Based on the consideration of the model selection requirements for model testing into the chatbot program, models such as GRU, Bi-GRU, 1D CNN and 1D CNN Transpose were selected.

(c)　　Determine the program evaluation method

There are several program methods that can be used, such as loss and accuracy. Based on the definition of accuracy from reference [35], accuracy is an attribute for an output value, which is described as $\Delta p_c$ deviation from the associated input value after going through mathematical operations. The information content of numeric entities can be observed accurately if they match the given references. The reference is the result of the derivative of the mathematical or physical system under consideration. The accuracy value represents the measurement for deviation from the reference.

In a previous study [33], the accuracy measurement method will be shown in the following equation to compare the performance of DNN and MLP in the QSAR model for the classification of acetylcholinesterase inhibitors:

$$accuracy = \frac{TP + TN}{TP + FP + TN + FN} \tag{16}$$

The following are the details of the components of Equation (16):

- True Positive (TP) means that if the value being tested is actually active, then the predicted value is also active.
- True Negative (TN) means that if the value being tested is actually not active, then the predicted value is also not active.
- False Positive (FP) means that if the actual value being tested is not active, then the predicted value is active.
- False Negative (FN) means that if the prediction value is active, then the actual value being tested is not active.

In evaluating MLP, a confusion matrix is used to obtain the value of the use of TP, FP, TN and FN (as discussed in detail in Equation (16)), which will produce an accuracy value in Equation (16). The illustrative form of the explanation above regarding the details of the components of Equation (16) will be shown in Table 1.

**Table 1.** Confusion matrix [36].

| Predicted Label | Actual Label | |
|---|---|---|
| | **Active** | **Inactive** |
| Active | True Positive (TP) | False Negative (FN) |
| Inactive | False Positive (FP) | True Negative (TN) |

The loss value can be obtained with several preparations [37]. They include observing *x*; therefore, the equation of the loss function can be stated as follows [33]:

$$L(\hat{y}, y) = How\ much\ difference\ from\ the\ true\ value\ of\ y \tag{17}$$

The following is used to calculate how close the classifier output ($\hat{y} = \sigma(w.x + b)$) is to the actual output (*y*, which is 0 or 1). The loss displayed in the sample chatbot program visualisation test must be based on a valid reference; therefore, it is important to clearly know the details of the form of loss that will be used as a program evaluation method.

*2.7. Data Identity*

Based on sources from reference [14], News Aggregator Dataset (data for November 2016) is used to test the classification model against predetermined categories. A collection of databases, domain theory and data generators derived from datasets obtained from the UCI Machine Learning Repository will be used in empirical analysis of learning algorithms for the machine learning community. The dataset can be accessed at https: //www.kaggle.com/uciml/news-aggregator-dataset (accessed on 30 June 2021). This dataset contains parameters by which one category can be affected by a sentence, namely,

ID is the numeric ID of the article, TITLE is the title of the article, URL is the URL of the article, PUBLISHER is the publisher of the article and CATEGORY is the category item. The meaning of each category in the data is t: science and technology; b: business.; m: health; e: entertainment. STORY is the alphanumeric ID of the news discussed in the article. HOSTNAME is the name of the host where the article is published. TIMESTAMP is the approximate time stamp of the publication of the article. The data shown in Table 2 is a detailed sample which is a form of input that will be processed into the chatbot program, based on the dataset used in this study.

**Table 2.** Sample details of chatbot data input [14].

| ID | Title | URL | Publisher | Category |
|---|---|---|---|---|
| 1 | Fed official says weather causing weak data should not slow down the taper | http://www.latimes.com/business/money/la-fi-mo-federal-reserve-plosser-stimulus-economy-20140310,0,1312750.story (accessed on 30 June 2021) | Los Angeles Times | b |
| 2 | The changing tapering speed seen from the high bar by Charles Plsser of Fed | http://www.livemint.com/Politics/H2EvwJSK2VE6OF7iK1g3PP/Feds-Charles-Plosser-sees-high-bar-for-change-in-pace-of-ta.html,Livemint,b,ddUyU0VZz0BRneMioxUPQVP6sIxvM (accessed on 30 June 2021) | Livemint | b |
| … | … | … | … | … |
| 51918 | Samsung Galaxy S5 available from April11 in India, starting at Rs. 51000 | http://www.livemint.com/Consumer/PsVzSYk54FGBNUUwh6eCAO/Samsung-Galaxy-S5-available-from-11-April-in-India-starting.html (accessed on 30 June 2021) | Livemint | t |
| 51919 | Government-imposed sanctions result in carriers jumping the gun on Galaxy S5 … | http://www.dailytech.com/South+Korean+Carriers+Begin+Sales+of+Galaxy+S5+Early+Samsung+Expresses+Regret/article34603c.htm (accessed on 30 June 2021) | DailyTech | t |
| … | … | … | … | … |
| 210749 | Jada's wife in New York City joins Will Smith for Fox Upfronts! | http://www.justjared.com/2005/14/12/will-smith-joins-wife-jada-in-new-york-city-for-fox-upfronts/ (accessed on 30 June 2021) | Just Jared | e |
| 210750 | Upfronts 2014: 'American Idol' gets little love at Fox presentation | http://www.latimes.com/entertainment/tv/showtracker/la-et-st-upfronts-2014-american-idol-gets-little-love-at-fox-presentation-20140512-story.html (accessed on 30 June 2021) | Los Angeles Times | e |
| … | … | … | … | … |
| 422933 | Surgeons to remove 4-year-old's rib to rebuild damaged throat - CBS 3 … | http://www.cbs3springfield.com/story/26378648/surgeons-removed-4-year-olds-rib-to-rebuild-his-damaged-throat (accessed on 30 June 2021) | WSHM-TV | m |

### 2.8. Creation of the Chatbot Program

Based on the chatbot programming flowchart in Figure 4, the following are detailed steps for creating the chatbot program by implementing the basic structure into the program.

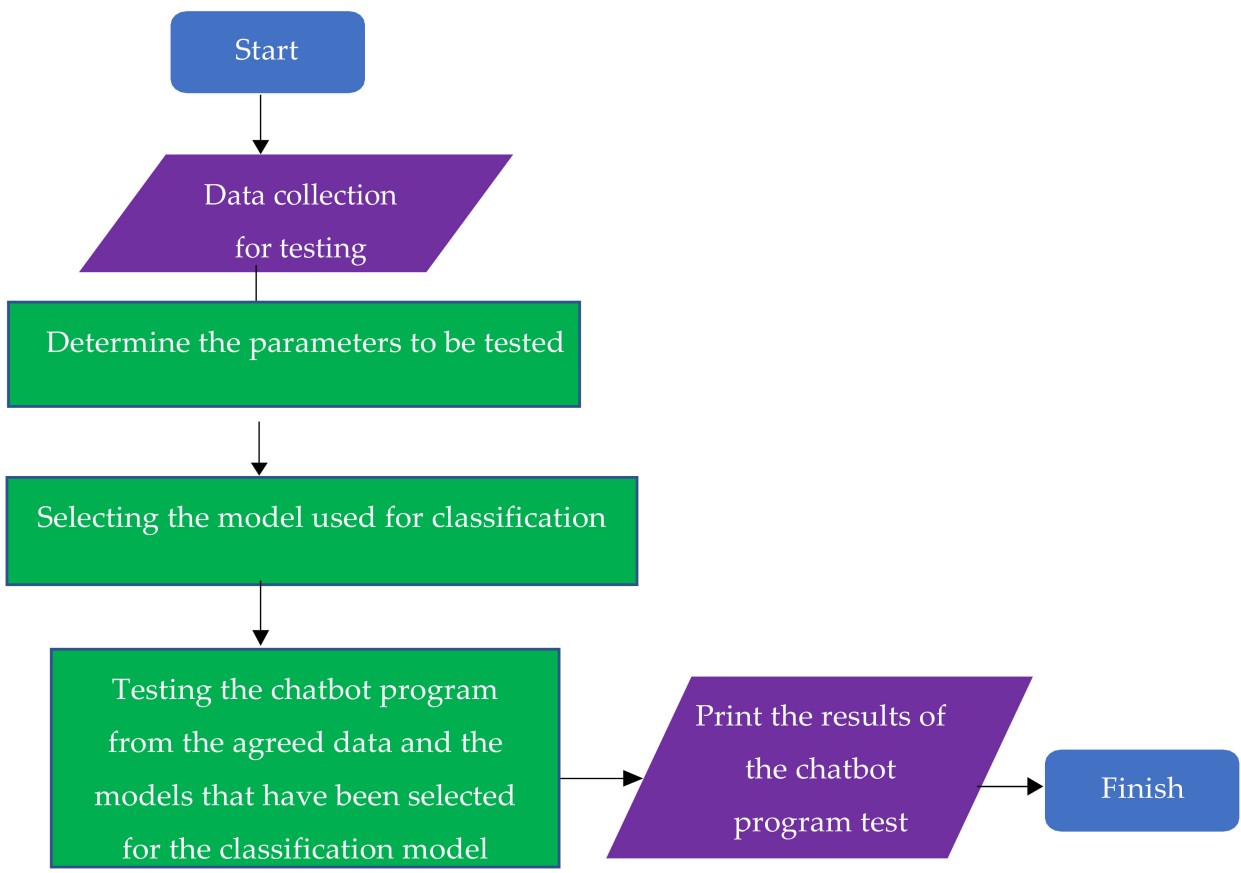

**Figure 4.** Chatbot programming flowchart.

1. Understanding how the model is applied to the program, in determining the best model, 4 models will be tested in the chatbot program, namely, GRU, Bi-GRU, 1D CNN and 1D CNN Transpose, which has 6 variation parameters in 36 different files.
2. Import the function to be used in the program.
3. Read files based on the location of the files to be used and determine the amount of data per category based on data from the dataset used.
4. Determine the number of categories that are used as references for dishuffles (in this program, adjust the number of categories in the dataset as much as 4). Randomly randomise the dataset; therefore, the data used in the program can be tested for its performance based on random data.
5. Perform the encoding stage on the 'CATEGORY' column, with the aim of changing the category data into four different matrix forms.
6. Determine the number of test_size to be tested (in this program, test_size = 0.20). Set the parameters to be tested (for example, it can be determined manually by the number of values for each parameter and this program has explained the variation of parameters in sub-chapter 3.3).
7. Print the form (X_train.shape, y_train.shape, X_test.shape, y_test.shape) of the data form to be tested into the program.
8. Build a model, print a summary of the shape of the model, fit the model and print the model test time on the model to be tested into the program; in step 6, images will be displayed based on the application of six different models into the program, which uses a variety of parameters according to step 6.
9. Print graphs of 'Graph of testing accuracy' and 'Graph of testing loss' that are the results of the model test used for the program; in step 6, images will be displayed based on the application of six different models into the program, which use various parameters according to step 6.

10. Conduct a phase 1 trial: sentences entered in [""], with the initial four sentences being tested sentences from the dataset, are directly entered in txt (in the trials in steps 10–12, one example from the test File 1 GRU Model will be shown).
11. Conduct a phase 2 trial, with the next four sentences; a sentence test that is made independently with the aim of finding whether the test results can be applied with a new sentence.
12. Conduct a phase 3 trial, with the next four sentences; a sentence test using the input dialog box as a form of a chatbot with sentences made independently.

## 3. Results

In this chapter, computer and software specifications will be discussed as the results of several runs of program files that have been tested through various amounts of parameter variations that have been adjusted to obtain the best accuracy value in the accuracy parameter.

### 3.1. Computer and Software Specifications

The research process uses the Python 3.8 programming language with the following computer and software specifications shown in Table 3.

**Table 3.** Computer and software specification table.

| Laptop Type | Lenovo Yoga Slim 7 14ITL05 |
| --- | --- |
| Processor | Intel Core i7-1165G7 Tiger Lake (2.8 GHz, 4 cores) |
| RAM | 16 GB |
| Graphic card | Intel Iris Xe Graphics |
| Storage | 1TB M.2 NVMe PCIe SSD |
| System of operating | Home Single Language 64-bit from Windows 10 |

### 3.2. Chatbot Program Test Results

Before going deeper, in 2021, the author tested the LSTM and BiLSTM models in a chatbot program. Here, Cornell Movie Dialog Corpus is used by the author, where it contains a dataset in the form of a corpus, which consists of a metadata-rich fictional collection extracted from movie scripts. Based on the experience of the dataset, various other types of models will be tested in this study [38]. In this research experimental setup, the proportion of detailed train data is 144,000, which is 80 percent of the total dataset, and the test data is 36,000, which is 20 percent of the total dataset.

The chatbot program was tested using various variations of the choice of the number of parameters and models, with the hope of obtaining the best results from the various program tests that have been carried out. As there are many files to be explained, the more specific details of the program test results can be seen in Tables 4–7. The following is an explanation of the parameter names used in the test program:

- n_most_common_words is the highest number of different words appearing from the data;
- test_size is the size of the test data to be processed;
- batch_size is the size of the batch that will be applied to the program, the effect of the size of the batch will affect the performance of the program [39];
- Epochs are how many iterations will be tested by adjusting the other parameter pairs [39];
- embedded_dim is the number of dimensions of the embedded vector that will be applied to the program [40];
- accuracy is an attribute for an output value based on [34], which, in this study, is used as a comparison between the program test results.

In Files one to six that use the GRU model with several variations of parameters to be tested in the program, the details of the parameter variations and program test results are shown in Table 4.

**Table 4.** Chatbot program test results (Files 1 to 6).

| Parameter Type | File 1 (GRU Model) | File 2 (GRU Model) | File 3 (GRU Model) | File 4 (GRU Model) | File 5 (GRU Model) | File 6 (GRU Model) |
|---|---|---|---|---|---|---|
| n_most_common _words | 4000 | 4000 | 4000 | 4000 | 4000 | 4000 |
| test_size | 100 | 100 | 100 | 100 | 100 | 100 |
| batch_size | 64 | 64 | 64 | 64 | 64 | 64 |
| Epochs | 10 | 10 | 10 | 20 | 20 | 20 |
| emb_dim | 64 | 128 | 256 | 64 | 128 | 256 |
| batch_size | 128 | 256 | 512 | 128 | 256 | 512 |
| accuracy | 0.9574 | 0.9570 | 0.9516 | 0.9845 | 0.9815 | 0.9814 |

In Files 7 to 12 that use the Bi-GRU model with several variations of parameters to be tested in the program, the details of the parameter variations and program test results are shown in Table 5.

**Table 5.** Chatbot program test results (Files 7 to 12).

| Parameter Type | File 7 (Bi-GRU Model) | File 8 (Bi-GRU Model) | File 9 (Bi-GRU Model) | File 10 (Bi-GRU Model) | File 11 (Bi-GRU Model) | File 12 (Bi-GRU Model) |
|---|---|---|---|---|---|---|
| n_most_common _words | 4000 | 4000 | 4000 | 4000 | 4000 | 4000 |
| test_size | 100 | 100 | 100 | 100 | 100 | 100 |
| batch_size | 64 | 64 | 64 | 64 | 64 | 64 |
| Epochs | 10 | 10 | 10 | 20 | 20 | 20 |
| emb_dim | 64 | 128 | 256 | 64 | 128 | 256 |
| batch_size | 128 | 256 | 512 | 128 | 256 | 512 |
| accuracy | 0.9585 | 0.9583 | 0.9535 | 0.9850 | 0.9844 | 0.9815 |

In Files 13 to 18 that use the 1D CNN model with several variations of parameters to be tested in the program, the details of the parameter variations and program test results are shown in Table 6.

**Table 6.** Chatbot program test results (Files 13 to 18).

| Parameter Type | File 13 (1D CNN Model) | File 14 (1D CNN Model) | File 15 (1D CNN Model) | File 16 (1D CNN Model) | File 17 (1D CNN Model) | File 18 (1D CNN Model) |
|---|---|---|---|---|---|---|
| n_most_common _words | 4000 | 4000 | 4000 | 4000 | 4000 | 4000 |
| test_size | 100 | 100 | 100 | 100 | 100 | 100 |
| batch_size | 64 | 64 | 64 | 64 | 64 | 64 |
| Epochs | 10 | 10 | 10 | 20 | 20 | 20 |
| emb_dim | 64 | 128 | 256 | 64 | 128 | 256 |
| batch_size | 128 | 256 | 512 | 128 | 256 | 512 |
| accuracy | 0.9691 | 0.9632 | 0.9580 | 0.9913 | 0.9910 | 0.9906 |

In Files 19 to 24 that use the 1D CNN Transpose model with several variations of parameters to be tested in the program, the details of the parameter variations and program test results are shown in Table 7.

**Table 7.** Chatbot program test results (Files 19 to 24).

| Parameter Type | File 19 (1D CNN Transpose Model) | File 20 (1D CNN Transpose Model) | File 21 (1D CNN Transpose Model) | File 22 (1D CNN Transpose Model) | File 23 (1D CNN Transpose Model) | File 24 (1D CNN Transpose Model) |
|---|---|---|---|---|---|---|
| n_most_common_words | 4000 | 4000 | 4000 | 4000 | 4000 | 4000 |
| test_size | 100 | 100 | 100 | 100 | 100 | 100 |
| batch_size | 64 | 64 | 64 | 64 | 64 | 64 |
| Epochs | 10 | 10 | 10 | 20 | 20 | 20 |
| emb_dim | 64 | 128 | 256 | 64 | 128 | 256 |
| batch_size | 128 | 256 | 512 | 128 | 256 | 512 |
| accuracy | 0.9693 | 0.9644 | 0.9590 | 0.9919 | 0.9917 | 0.9915 |

Figure 5 shows this accuracy comparison diagram of all the chatbot programs based on the 24 tested files that will be displayed in the form of a diagram.

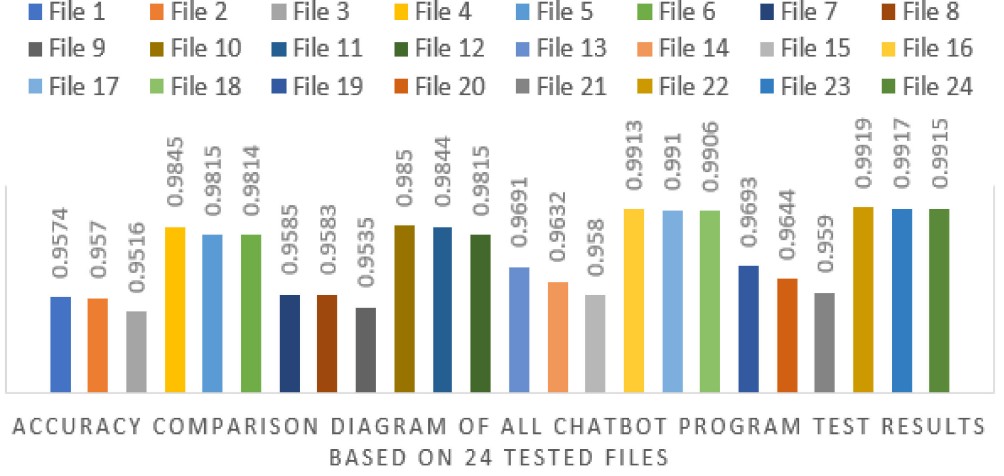

**Figure 5.** Accuracy comparison diagram of all chatbot program test results based on 24 tested files.

Based on the detailed various program trials from Table 4 to Table 7, and the accuracy comparison diagram in Figure 5, it was found that File 22 with the 1D CNN Transpose model and the details of the parameter variations, as in the table, produce an accuracy of 0.9919. We believe that the results of this study are not in overfitting conditions (when the training accuracy is very good, but the test accuracy is not good); in this study, we tested based on the accuracy obtained from testing the test data, which we have tested from various test variations from datasets using various types of data variations and retested with the arrangement data entered by the user.

## 4. Discussion

After obtaining the results of the program testing, the next step is to evaluate the results by verifying the results with references, which aims to find out whether the results obtained are in accordance with the references used as references in the application of the four models to be tested into the program. The four chatbots are GRU, Bi-GRU, 1D

CNN and 1D CNN Transpose. The use of the best-tested model is based on Table 7, then a discussion of various comparisons of program test results in various references will be discussed in this discussion section. In the research proposed for the use of the 1D CNN Transpose model in the chatbot program, the reference source [34] has an average value of structural similarity, which, in the image object, calculates the value of similarity over the predicted value of the CNN model. The transpose that can be adjusted for the number of dimensions is in accordance with the illustration in Figure 3 based on [33], with an average value of 0.9213. Compared to using the same dataset in reference [20], using the Recurrent Graph Neural Network or R-GNN(-replygraph) obtains the highest accuracy of 0.7057. The accuracy of the chatbot research that uses text dialog data, the best results obtained using the compressed BiLSTM model have an accuracy of 78% [41]. As a comparison from the application of the 1D CNN Transpose model to the chatbot program in this study, according to Table 7, the test results produced have an accuracy value of 0.9919, which is indeed better than the entire reference referenced.

Based on the program testing, the results of the chatbot program trial are expected to be able to apply a model that can produce even better test results in the future. It is hoped that in the future, chatbots can be integrated with image data, so that the results of the chatbot will be more comprehensive. In addition, further development can be performed using transfer learning and NLP for out-of-context questions.

In the future, for research development plans, we have plans to increase the amount of data to be tested in the program, as well as develop models that follow technological developments, with the aim that researchers obtain better program test results [42].

The following is the documentation of the visualisation of how the chatbot program works, along with the test sentences in Figures 6–8.

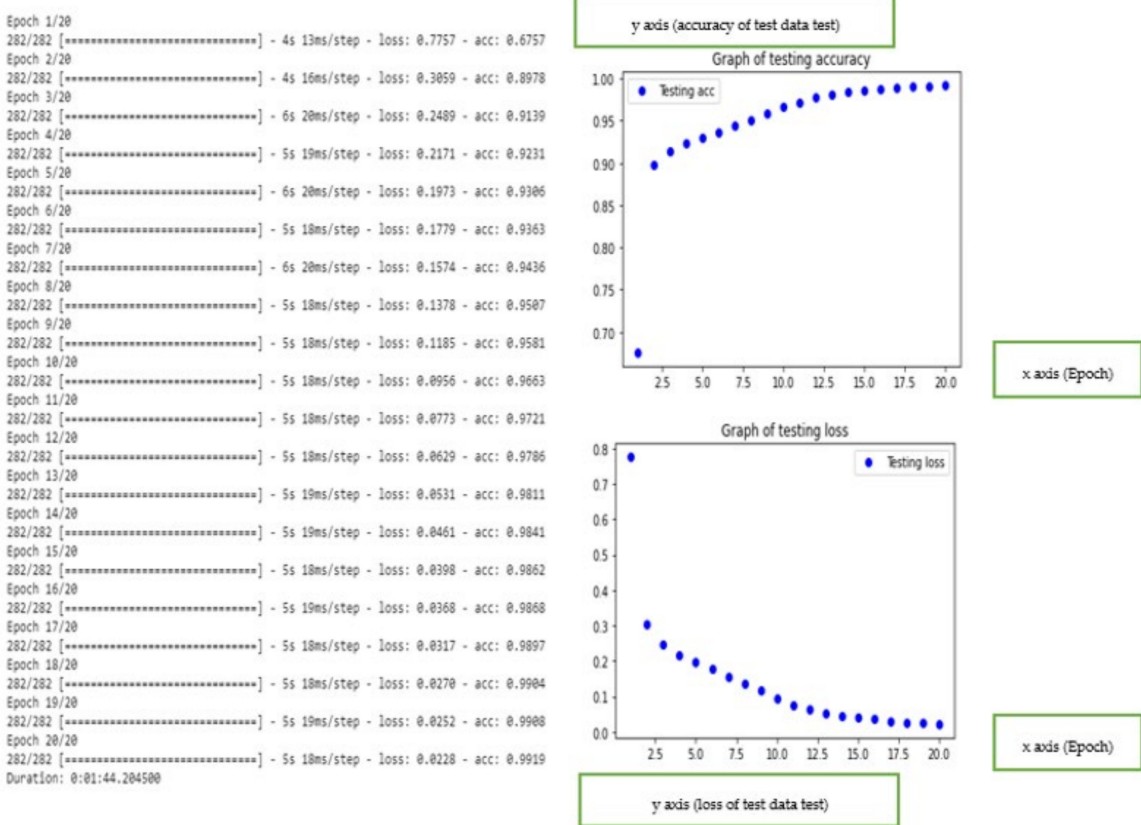

**Figure 6.** Details of accuracy and loss values in program testing, graph of testing accuracy and graph of testing loss (one example of the test results of the File 22 program with program details as shown in Table 7).

```
In [14]:  ▶ txt = ["Kids not the only ones guilty of too much screen time"]
            seq = tokenizer.texts_to_sequences(txt)
            padded = pad_sequences(seq, maxlen=max_len)
            pred = model.predict(padded)
            labels = ['entertainment', 'bussiness', 'science/tech', 'health']
            print(pred, labels[np.argmax(pred)])

            [[9.9518508e-01 8.2134790e-08 6.5041744e-13 4.8148655e-03]] entertainment
```

```
In [15]:  ▶ txt = ["Unicredit posts 14 bn euro loss for 2013"]
            seq = tokenizer.texts_to_sequences(txt)
            padded = pad_sequences(seq, maxlen=max_len)
            pred = model.predict(padded)
            labels = ['entertainment', 'bussiness', 'science/tech', 'health']
            print(pred, labels[np.argmax(pred)])

            [[1.27893225e-14 1.00000000e+00 1.86382377e-13 1.11036527e-18]] bussiness
```

```
In [16]:  ▶ txt = ["Taylor Swift reclaims 'top earner' crown"]
            seq = tokenizer.texts_to_sequences(txt)
            padded = pad_sequences(seq, maxlen=max_len)
            pred = model.predict(padded)
            labels = ['entertainment', 'bussiness', 'science/tech', 'health']
            print(pred, labels[np.argmax(pred)])

            [[9.9992895e-01 5.3091355e-08 7.0874201e-05 2.9296242e-08]] entertainment
```

```
In [17]:  ▶ txt = ["Drug company denies dying child access to life-saving medicine"]
            seq = tokenizer.texts_to_sequences(txt)
            padded = pad_sequences(seq, maxlen=max_len)
            pred = model.predict(padded)
            labels = ['entertainment', 'bussiness', 'science/tech', 'health']
            print(pred, labels[np.argmax(pred)])

            [[4.6512703e-14 2.6523732e-14 1.4671028e-22 1.0000000e+00]] health
```

**Figure 7.** Sentence testing in [" "], with a total of four initial sentences, sentence trials from the dataset were directly entered in txt (one example of the test results of the File 22 program with program details as shown in Table 7).

```
In [19]:  ▶ txt = ["Doctor give patient medicine with many option"]
            seq = tokenizer.texts_to_sequences(txt)
            padded = pad_sequences(seq, maxlen=max_len)
            pred = model.predict(padded)
            labels = ['entertainment', 'bussiness', 'science/tech', 'health']
            print(pred, labels[np.argmax(pred)])

            [[8.4023297e-16 1.5924829e-14 4.5035306e-19 1.0000000e+00]] health
```

```
In [20]:  ▶ txt = ["Company sell product with high value in few month"]
            seq = tokenizer.texts_to_sequences(txt)
            padded = pad_sequences(seq, maxlen=max_len)
            pred = model.predict(padded)
            labels = ['entertainment', 'bussiness', 'science/tech', 'health']
            print(pred, labels[np.argmax(pred)])

            [[1.2626437e-06 9.9738950e-01 1.2630695e-03 1.3461412e-03]] bussiness
```

```
In [21]:  ▶ txt = ["Autonomous mobile is used in the modern era"]
            seq = tokenizer.texts_to_sequences(txt)
            padded = pad_sequences(seq, maxlen=max_len)
            pred = model.predict(padded)
            labels = ['entertainment', 'bussiness', 'science/tech', 'health']
            print(pred, labels[np.argmax(pred)])

            [[2.6882390e-02 6.3593929e-05 9.7305250e-01 1.4979496e-06]] science/tech
```

```
In [22]:  ▶ txt = ["Their performance last night is adored by audience"]
            seq = tokenizer.texts_to_sequences(txt)
            padded = pad_sequences(seq, maxlen=max_len)
            pred = model.predict(padded)
            labels = ['entertainment', 'bussiness', 'science/tech', 'health']
            print(pred, labels[np.argmax(pred)])

            [[9.9999118e-01 1.5997598e-07 5.5200239e-06 3.1627387e-06]] entertainment
```

**Figure 8.** Sentence testing in [" "], the sentence test in [" "], with the next 4 sentences, a sentence test that is made independently is carried out with the aim of whether the test results can be applied to the new sentence (one example of the test results of the File 22 program with program details as shown in Table 7).

## 5. Conclusions

Based on the test results of the chatbot program that has been built using the application of various models and parameter variations from 24 files that have been tested, it can be stated that the results of the analysis of the use of the best model using the 1D CNN Transpose model in the chatbot program are as shown in Table 7 of the program test results and the resulting accuracy value is 0.9919.

**Author Contributions:** All authors have contributed equally. P.A. has contributed in terms of writing from research documentation, program development, to revisions from reviewers. A.B. has contributed to the financing of publication costs, guided the writing of the draft, and provided advice on the reviewer's revision response. R.A.B. has contributed in checking the meaning of writing research documentation, guiding in drafting, as well as providing suggestions on the reviewer's response to revisions. All authors have read and agreed to the published version of the manuscript.

**Funding:** Part of this research was supported by PUTI KI 2Q2 2020 research grant from Universitas Indonesia with contract number NKB-778/UN2.RST/HKP.05.00/2020.

**Institutional Review Board Statement:** Ethical review and approval were waived for this study, due to aims to develop better research in the field of computer science in the future.

**Informed Consent Statement:** Informed consent was obtained from all subjects involved in the study.

**Data Availability Statement:** Publicly available datasets were analysed in this study. This data can be found here: https://www.kaggle.com/uciml/news-aggregator-dataset.

**Acknowledgments:** To members of the Laboratory of Bioinformatics and Advanced Computing (BACL), Department of Mathematics and Data Science (DSC), Faculty of Mathematics and Natural Sciences, University of Indonesia, the authors are grateful for your support.

**Conflicts of Interest:** The authors declare that there is no conflict of interest, and the funders were not involved in the research design, data collection and analysis or interpretation of the data. In addition, the funder was not involved in the decision to publish this manuscript.

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
