# Peer review of "Comparative Analysis of Performance between Multimodal Implementation of Chatbot Based on News Classification Data Using Categories"

_electronics, doi:10.3390/electronics10212696_

Round 1
Reviewer 1 Report
1. In introduction, research question and research goals should be clearly stated.
2. Literature survey should be expanded with more recent publications dealing with chatbots. Include the following recent MDPI Electronics publications at least:
https://www.mdpi.com/1999-5903/12/7/109
https://www.mdpi.com/2079-9292/10/18/2300
https://www.mdpi.com/2079-9292/10/6/666
3. Authors should improve the quality of the Figures. For example, Figure 1 is a bit blurry and it should be given in higher resolution. Same applies for other figures.
4. In equations, not all parameters have been explained and described at the place they were introduced (Sigma in Eq. 4 for example, although it has been described later).
5. Tables 4-7 header cells should be fixed, because text is broken strangely.
6. State clearly the experimental setup - the amount of data used for training, for testing etc.
7. Comparative analysis with similar state of the art approaches that were tested on the same dataset should be provided.
8. English language should be improved. For instance, consider the sentence:
In text mining, they use a lot of NLP in processing text and speech [27].
It can be rephrased in the following way.
Text mining extensively use NLP in processing text and speech.
Or, the sentence:
In the processing of machine learning tasks and natural language, the problem of text classification will often be used, so it must be watched.
It is not clear what authors meant to say, so it should be rephrased.
Authors should extensively check the complete manuscript for such strange sentence structures.
Author Response
Here I attach proof of submission of the revised manuscript and responses from reviewers' comments based on discussions with other authors, grammatical consultations and further literature study. I hope there is a response from the reviewer regarding the revision regarding whether or not the revision was accepted, because it relates to my report responsibilities in the grant report.
Thank you so much for being so cooperative.

Reviewer 2 Report
Sometimes it is hard to go through the text, though I am not a native speaker. Hence I suggest paying additional attention to improving English and text style and consistency.
The motivation and objectives of the paper should be indicated more clearly.
The introduction looks like a compilation from the different sources without good liaison and consistency. E.g., “The impact in daily activities has been given by artificial intelligence (AI) through the design and evaluation of applications and also the sophisticated devices that can perform various functions. Chatbot is a form of intelligence based on AI” (lines 32-34), and again in lines 50-54.
I suggest the introduction be rewritten, keeping in mind the main point the authors want to say to a reader. Moreover, the entire text looks gathered from the parts without the general direction. I suggest the paper be thoroughly rewritten to be a single text, easy to read and understand.
Please describe the experiments, viz. testing the chatbot program, in more detail. This part is essential for the research paper. However, it is almost totally omitted in this article. The size of the dataset, data preprocessing, if any, need to be described in the paper. How did the authors deal with the overfitting problem? All these important questions remain open and are not discussed.
Line 396: “When compared to the results of trials on research referrals on chatbots …” So, where are these results? Why are they neither described nor even cited? The comparison with [31] is incorrect because paper [31] deals with image resolution processing and not a chatbot.
Line 389: “Then it will be discussed in this sub-chapter whether it can be proven true.” Where is this discussion? So, can the obtained results be proved? The question, in my opinion, remains open.
The authors state they have reached a classification accuracy value of 0.9919. However, it is unclear for which data size it was obtained, whether there was overfitting or not, how reproducible the obtained results are, etc.
To summarize: the paper needs substantial revision both in its content and language.
Some other comments:
- I do not understand how and for what purpose the database set, domain theory and data generator datasets generated by the UCI Machine Learning Repository (lines 102-105) were used in this paper.
- The values that appear in Fig. 1 should be explained in the text. The gates mentioned in Eq. (1)-(5) do not appear in Fig. 1, so the GRU model and process remain unclear for a reader unfamiliar with them.
- All the parameters mentioned in Tables 4-7 (batch_size, emb_dim, etc.) should be explained.
- Table 8 contains no new information comparing to Table 7.
Style and language.
- A lot of sentences and paragraphs can be confusing to a reader. The examples could be, but are not limited to, the following:
- The lines 85-90 should be removed, I guess.
- In which way do the lines 75-78 relate to the previous text? Why are they placed here?
- Lines 127-128: “NLP and supervised machine learning tasks are used for text classification problems [17].” Why is this sentence placed here? How is it connected with previous and following sentences?
- Lines 172-174: “Parallel work based on parallel computing is a type of computing that can perform various process calculations that are carried out at the same time [23].” Is this sentence needed and needed precisely in this place?
- Line 185: “In the processing of machine learning tasks and natural language, the problem of text classification will often be used, so it must be watched” Why are you speaking about it here, in line 185?
- Line 189: “The trained Words2Vec model …” Have you ever mentioned this model before in the text? What is it? Why is it said here?
- Line 194-194: “Initially, image classification and computer vision problems were developed by CNN. However, many have now switched to NLP.” Do I understand well that natural language processing is now used for image classification and computer vision instead of convolutional neural networks?
- Lines 195-197: “Where the one-dimensional (ID) array text representation in NLP tasks because the form of text is one-dimensional data, it is needed when CNN is implemented in text instead of images.” Sorry, I do not understand this sentence at all.
- Lines 257-263: What is this discussion about? How does it influence the paper and its results?
- What is the purpose of using (17) as an equation? It is not an equation. You can describe it with words, as you did.
- Lines 296, 297, 298: Why are they so different in formatting from the previous text? They are the continuation of the last sentence, cut by the table, without any reference to this table. Please re-format this part of the text.
- Lines 338-340: Terrible English. Very hard to read.
- Lines 353-356: “After testing the chatbot program by changing different amounts on certain parameters, as well as several models that have been selected, with the hope of getting various results from the variation of parameters that have been tested in the program, so that in the end it can be seen, the best parameter pair among several program tests.” Improve English, please.
Author Response

(The authors gave the same response as above.)

Reviewer 3 Report
What I would like to see is some concrete examples of the chatbot in actual operation.
Author Response

(The authors gave the same response as above.)

Round 2
Reviewer 1 Report
Authors have addressed all the issues from the previous round of review, and the paper is in satisfactory state.
Author Response
Thank you for the quick revision feedback. In the following, we have made a minor revision of the suggestions given. I hope the reviewer can provide a review report on the results of this revision.

Reviewer 2 Report
The paper has been substantially improved. The research part has been clarified. The main remaining issue, in my opinion, is English. English of the paper is much better compared to the initial version; however, it still could be improved.
General remark: try to avoid very long sentences. This would improve the ease of reading of the article. Some examples are as follows:
- Lines 272-276: one long sentence.
- Lines 446-449 contain one long sentence, where the words "program test results" are repeated five times.
Minor comments:
Lines 93-96: so, what is the research question of the paper: the limited number of hours of customer service?
All tables should be mentioned in the text. Tables 1 and 2 are not referenced in the text.
Lines 320-334: Do not break a sentence with a table.
Line 328: ?
Line 399: train data -> test data
Line 473: do not refer to a figure as a continuation of the sentence with a colon. Mention the figures directly.
Author Response

(The authors gave the same response as above.)

Reviewer 3 Report
Accept
Author Response

(The authors gave the same response as above.)
